# AUGMENTING SELF-ATTENTION WITH PERSISTENT MEMORY

## ABSTRACT

Transformer networks have lead to important progress in language modeling and machine translation. These models include two consecutive modules, a feed-forward layer and a self-attention layer. The latter allows the network to capture long term dependencies and are often regarded as the key ingredient in the success of Transformers. Building upon this intuition, we propose a new model that solely consists of attention layers. More precisely, we augment the self-attention layers with persistent memory vectors that play a similar role as the feed-forward layer. Thanks to these vectors, we can remove the feed-forward layer without degrading the performance of a transformer. Our evaluation shows the benefits brought by our model on standard character and word level language modeling benchmarks.

## 1    INTRODUCTION

Transformer networks (Vaswani et al., 2017) are sequence models that rely on the attention mechanism (Bahdanau et al., 2015) to capture long term dependencies. Since their introduction in the context of machine translation, they have been applied to many natural language processing tasks, such as language modeling (Al-Rfou et al., 2019) or sentence representation (Devlin et al., 2019). On most of them, they are now surpassing the former state-of-the-art models based on recurrent (Hochreiter & Schmidhuber, 1997) or convolutional networks (Dauphin et al., 2017). At their core, transformers use a self-attention layer that forms a representation of the current input by gathering the most relevant information from its context. This layer is repeated along the network depth, allowing for information to flow for long distances and to form rich sequence representations. The self-attention mechanism is often considered as the key component of their success and many have worked on improving transformers by increasing the size of the context captured by those layers (Wu et al., 2019; Dai et al., 2019; Sukhbaatar et al., 2019).

However, self-attention layers are not the only component of transformer networks and they do not explain the effectiveness of transformers by themselves. Each of these layers is followed by a feedforward layer. These feedforward layers contain most of the parameters of the model. This suggests that their role is probably as important as the self-attention mechanism. In fact, the transformer layer, i.e., the sequence of self-attention and feedforward sublayers, should be regarded as a single mechanism that gathers information from the context and transforms it into a rich representation. Having such two different layer types of at the core makes Transformer models harder to analyse and understand. In particular, there are not many works exploring the properties of feedforward layers.

In this work, we simplify the transformer architecture by revisiting its mechanism, while keeping its properties. We introduce a new layer that merges the self-attention and feedforward sublayers into a single unified attention layer, as illustrated in Figure 1. As opposed to the two-step mechanism of the transformer layer, it directly builds its representation from the context and a persistent memory block without going through a feedforward transformation. The additional persistent memory block stores, in the form of key-value vectors, information that does not depend on the context. In terms of parameters, these persistent key-value vectors replace the feedforward sublayer. This modification dramatically simplifies the structure of the network with no loss of performance.

We evaluate the resulting architecture on standard word level and character level language modeling benchmarks and report performances that are competitive with transformers.

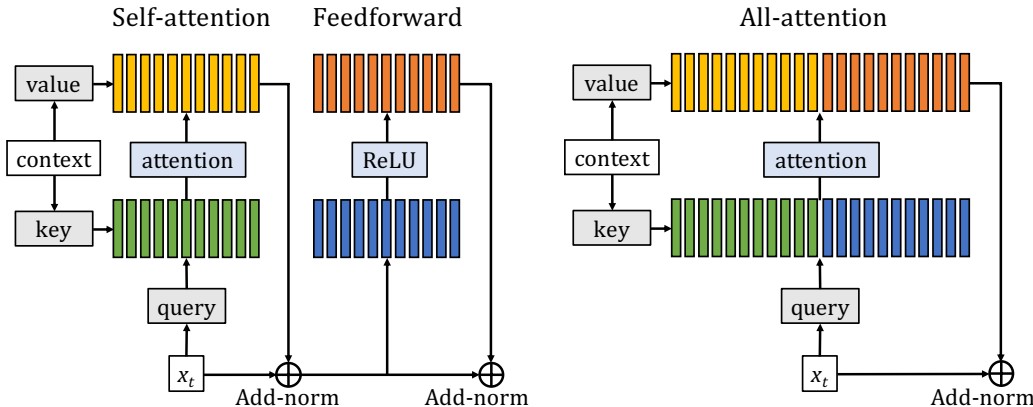

Figure 1: On the left panel, the standard transformer layer is composed of a self-attention sublayer followed by a feedforward sublayer. On the right panel, our all-attention layer merges the weights of the feedforward sublayer with the self-attention sublayer. We represent both models in the case of a single head, but in the general case, both the self-attention sublayer and our all-attention layers have multiple heads.

## 2 RELATED WORK

**Neural language modeling.** Different network architectures have been proposed for language modeling, such as feed-forward networks (Bengio et al., 2003a), recurrent networks (Mikolov et al., 2010), gated convolutional networks (Dauphin et al., 2017) and transformer networks (Vaswani et al., 2017). Of particular interest, Al-Rfou et al. (2019) apply deep transformers to character level language modeling. Dai et al. (2019) introduces a caching mechanism, relying on the relative position embeddings from Shaw et al. (2018), which makes inference in these models much more efficient for unbounded sequences. More recently, Sukhbaatar et al. (2019) add a learnable self-attention span to extend the size of the context.

Word level language models deal with large vocabularies and computing the most probable word is computationally demanding. Solutions are to either replace the softmax loss with an approximation (Goodman, 2001; Morin & Bengio, 2005), to sample from the vocabulary during training (Bengio et al., 2003b; Jozefowicz et al., 2016) or to include subword units (Sennrich et al., 2016). A simple yet effective solution is to replace the loss by a hierarchical softmax designed to better take advantage of the GPU specificities (Grave et al., 2017a).

Finally, many works focus on the regularization of large language models. In particular, Zaremba et al. (2014) show that dropout (Srivastava et al., 2014) is effective for recurrent networks. More recently, Press & Wolf (2017) show that tying the embedding and classifier weights significantly improves generalization. Baevski & Auli (2019) further show that combining this regularization technique with the adaptive softmax of (Grave et al., 2017a) reduces the memory footprint of a transformer while improving its performance.

**Attention based models.** The attention mechanism was first introduced in the context of mixture of experts by Jordan & Jacobs (1994). It is only recently that Bahdanau et al. (2015) have shown their potential when used in neural networks in the context of machine translation. Since then, this mechanism is commonly incorporated within many models, with applications in natural language processing and computer vision, besides transformers. Sukhbaatar et al. (2015) apply the attention mechanism on the same sequence, i.e., the so-called self-attention, in an auto-regressive model called end-to-end memory network. They show their potential in the context of language modeling. Graves et al. (2014) use the attention mechanism for reading from and writing to internal memory for solving algorithmic tasks. Vinyals et al. (2015) combine this self-attention mechanism with a recurrent network to solve simple algorithmic problems. Later, Merity et al. (2017) show that these networks can be used as language models if combined with a cache mechanism (Grave et al., 2017b). The attention mechanism has been also applied to question answering (Miller et al., 2016) and image

captioning (Xu et al., 2015). Finally, Shazeer et al. (2017) uses the attention mechanism as a mixture of experts in a recurrent network.

## 3  TRANSFORMER LAYER

A transformer model is made of a stack of identical layers, called transformer layers. Each layer is composed of a multi-head self-attention sublayer followed by a feedforward sublayer. Each sublayer is also followed by an add-norm operation, i.e., a skip-connection (He et al., 2016), and layer normalization (Lei Ba et al., 2016). In this section, we review the structure of the transformer layer and refer the reader to Vaswani et al. (2017) for additional details of the overall model.

**Multi-head self-attention sublayer.**   A core mechanism of a transformer network is the multi-head self-attention layer, which consists of multiple attention heads applied in parallel. Each attention head applies the attention mechanism of Bahdanau et al. (2015) on an input sequence of vectors. More formally, given a sequence $\mathbf{x}_1, ..., \mathbf{x}_T$ of $d$-dimensional input vectors, each head applies two linear transformations to these vectors to form the key and value vectors:

$$\mathbf{k}_t = \mathbf{W}_k \mathbf{x}_t, \tag{1}$$
$$\mathbf{v}_t = \mathbf{W}_v \mathbf{x}_t, \tag{2}$$

where $\mathbf{W}_k$ and $\mathbf{W}_v$ are the "key" and "value" matrices of a size $d_h \times d$, where $d_h = d/H$ is the dimension of a head and $H$ is the number of heads. The key vectors are then used to compute a similarity score between an element $t$ of the input sequence and all the elements of its context $C_t$. The context can be, for instance, the elements of the sequence that precede $t$ in the case of language modeling, or the whole sequence in the encoder for machine translation. The similarity score between $t$ and an element $c$ of its context $C_t$ is defined as

$$s_{tc} = \mathbf{x}_t^\top \mathbf{W}_q^\top \left( \mathbf{k}_c + \mathbf{p}(t, c) \right), \tag{3}$$

where $\mathbf{W}_q \in \mathbb{R}^{d_h \times d}$ is the "query" matrix, and $\mathbf{p}(t, c)$ is a position encoding function. There are several ways to encode positions: fixed absolute (Vaswani et al., 2017), learned absolute (Al-Rfou et al., 2019), and learned relative (Sukhbaatar et al., 2015; Shaw et al., 2018). The relative position encoding function improves the efficiency for unbounded sequences, making them useful for language modeling (Dai et al., 2019). In this paper, we thus use the relative position encoding defined as $\mathbf{p}(t, c) = \mathbf{u}_{t-c}$, where $\mathbf{u}_i$ are position embeddings learned during training. The head then outputs a vector $\mathbf{y}_t$ by taking the average of the context representations weighted by attention weights $a_{tc}$ obtained by applying a softmax function to the similarity scores:

$$\mathbf{y}_t = \sum_{c \in C_t} a_{tc} \left( \mathbf{v}_c + \mathbf{p}(t, c) \right) \quad \text{and} \quad a_{tc} = \frac{\exp\left( s_{tc}/\sqrt{d_h} \right)}{\sum_{i \in C_t} \exp\left( s_{ti}/\sqrt{d_h} \right)}. \tag{4}$$

Note that one can use different position encoding functions for the key and value sides. Finally, the outputs from the different heads are concatenated for each timestep $t$ and multiplied by the $d \times d$ "output" matrix $\mathbf{W}_o$. The final output of this sublayer is thus a sequence of $T$ vectors of dimension $d$.

**Feedforward sublayer.**   The second element of a transformer layer is a fully connected feedforward layer. This sublayer is applied to each position $t$ in the input sequence independently, and consists of two affine transformations with a pointwise non-linear function in between:

$$\mathrm{FF}(\mathbf{x}_t) = \mathbf{U}\, \sigma \left( \mathbf{V} \mathbf{x}_t + \mathbf{b} \right) + \mathbf{c}, \tag{5}$$

where $\sigma(x) = \max(0, x)$ is the ReLU activation function; $\mathbf{V}$ and $\mathbf{U}$ are matrices of dimension $d \times d_f$ and $d_f \times d$ respectively; $\mathbf{b}$ and $\mathbf{c}$ are the bias terms. Typically, $d_f$ is set to be 4 times larger than $d$.

**Add-norm.**   Both the multi-head self-attention and the feed-forward layer are followed by an add-norm operation. This transformation is simply a residual connection (He et al., 2016) followed by layer normalization (Lei Ba et al., 2016). The layer normalization computes the average and standard deviation of the output activations of a given sublayer and normalizes them accordingly. This guarantees that the input $\mathbf{y}_t$ of the following sublayer is well conditioned, i.e., that $\mathbf{y}_t^T \mathbf{1} = 0$ and $\mathbf{y}_t^T \mathbf{y}_t = \sqrt{d}$. More precisely, the AddNorm operation is defined as:

$$\mathrm{AddNorm}(\mathbf{x}_t) = \mathrm{LayerNorm}(\mathbf{x}_t + \mathrm{Sublayer}(\mathbf{x}_t)), \tag{6}$$

where Sublayer is either a multi-head self-attention or a feedforward sublayer.

**Transformer layer.** The overall transformer layer has the following set of equations:

$$\mathbf{z}_t \;=\; \text{AddNorm}(\text{MultiHead}(\mathbf{x}_t)), \tag{7}$$

$$\mathbf{y}_t \;=\; \text{AddNorm}(\text{FF}(\mathbf{z}_t)), \tag{8}$$

where `MultiHead` is the multi-head self-attention sublayer. This is shown on the left panel of Fig. 1.

## 4 OUR APPROACH

In this section, we first show that a feedforward sublayer can be viewed as an attention layer. Then, we take advantage of this interpretation of a feedforward model to concatenate it with the self-attention layer, forming a novel layer that relies solely on a multi-head attention layer without the need for a feedforward sublayer.

### 4.1 FEEDFORWARD SUBLAYER AS AN ATTENTION LAYER

We transform the feedforward sublayer into an attention layer by replacing the `ReLU` non-linear function in Eq. 5 by a `Softmax` function and removing the biases:

$$\mathbf{y}_t = \mathbf{U}\text{Softmax}(\mathbf{V}\mathbf{x}_t) = \sum_{i=1}^{d_f} a_{ti}\mathbf{U}_{*,i}. \tag{9}$$

Here we use notations $\mathbf{U}_{*,i}$ and $\mathbf{V}_{i,*}$ to denote column and row vectors respectively. The activation $a_{ti}$ is thus the attention weight computed with $\mathbf{V}_{i,*}$ and $\mathbf{x}_t$. The vectors $\mathbf{x}_t$, $\mathbf{V}_{i,*}$ and $\mathbf{U}_{*,i}$ are equivalent to the query, key and value vectors respectively. The Eq. 9 is also equivalent to the self-attention sublayer of Eq. 3-4 with the context vectors $\mathbf{k}_t$, $\mathbf{v}_t$ set to zero and the vectors $\mathbf{V}_{i,*}$ and $\mathbf{U}_{*,i}$ are used as key and value side position embeddings respectively. This allows for a similar implementation for the feedforward and the self-attention sublayers, and opens the possibility of merging them into a single layer.

### 4.2 PERSISTENT MEMORY AUGMENTED SELF-ATTENTION LAYER

Here we propose a single attention layer that can replace both self-attention and feedforward layers in Transformers, which we call *all-attention* layer. Our layer applies the attention mechanism simultaneously on the sequence of input vectors, as in the standard self-attention layer, and on a set of vectors not conditioned on the input. These vectors are added to capture information that does not depend on the immediate context, like general knowledge about the task. They are shared across the data and, in some sense, forms a persistent memory similar to the feedforward layer. Therefore we call them *persistent vectors*. More precisely, the persistent vectors are a set of $N$ pairs of key-value vectors, respectively stacked in two $d_h \times N$ dimensional matrices $\mathbf{M}_k$ and $\mathbf{M}_v$. As discussed in Section 4.1, $\mathbf{M}_k$ and $\mathbf{M}_v$ can be interpreted as $\mathbf{V}$ and $\mathbf{U}$ of a feedforward sublayer.

These persistent vectors are simply added to the pool of key and value vectors conditioned on the input:

$$[\mathbf{k}_1, \ldots, \mathbf{k}_{T+N}] \;=\; \text{Concat}\left([\mathbf{W}_k\mathbf{x}_1, \ldots, \mathbf{W}_k\mathbf{x}_T], \mathbf{M}_k\right), \tag{10}$$

$$[\mathbf{v}_1, \ldots, \mathbf{v}_{T+N}] \;=\; \text{Concat}\left([\mathbf{W}_v\mathbf{x}_1, \ldots, \mathbf{W}_v\mathbf{x}_T], \mathbf{M}_v\right). \tag{11}$$

Let us denote by $C_t^+$ the concatenation of the context $C_t$ and the indices corresponding to the $N$ persistent vectors. The similarity score between an element $t$ of the input sequence and an element $c$ of its extended context $C_t^+$ is computed the same way as in Eq. (3), i.e.:

$$s_{tc} = \mathbf{x}_t^\top \mathbf{W}_q^\top \left(\mathbf{k}_c + \mathbf{p}(t, c)\right), \tag{12}$$

where the position encoding corresponding to a persistent vector is equal to zero. The all-attention then outputs a vector $\mathbf{y_t}$ with the same attention function as in Eq. (4), i.e.,

$$\mathbf{y}_t = \sum_{c \in C_t^+} a_{tc}\left(\mathbf{v}_c + \mathbf{p}(t, c)\right) \quad \text{and} \quad a_{tc} = \frac{\exp\left(s_{tc}/\sqrt{d_h}\right)}{\sum\limits_{i \in C_t^+} \exp\left(s_{ti}/\sqrt{d_h}\right)}. \tag{13}$$

As with a self-attention sublayer, an all-attention layer can have multiple heads, where outputs from the different heads are concatenated for each timestep $t$ and multiplied $\mathbf{W}_o$. Note that persistent vectors are not shared between heads. Our overall layer is then simply this new `MultiHeadAllAttn` sublayer followed by the `AddNorm` operation as defined in Eq. (6), i.e.,

$$\mathbf{y}_t = \text{AddNorm}(\text{MultiHeadAllAttn}(\mathbf{x}_t)). \tag{14}$$

The right panel of Fig. 1 summarize the all-attention layer in the case of a single head: we remove the feedforward sublayer and add unconditioned persistent vectors to the self-attention sublayer. While the persistent vectors are directly comparable to a feedforward sublayer in the case of a single head, a multi-head version is more comparable to multiple small feedforward layers working in parallel. If there are as many persistent vectors as the ReLU units, an all-attention layer has the same number of parameters as the standard transformer layer regardless of the number of heads (ignoring bias terms).

Note that using attention mechanism to address unconditioned persistent vectors has been previously proposed in the context of question answering with knowledge bases (Miller et al., 2016).

### 4.3 LANGUAGE MODELING

Language modeling is the problem of assigning a probability to a sequence of tokens $(w_1, \ldots, w_T)$:

$$P(w_1, \ldots, w_T) = \prod_{t=1}^{T} P(w_t \mid w_{t-1}, \ldots, w_1).$$

In this paper, we focus on tokens that are either words or characters. Language modeling has been dominated by neural networks with models either based on feedforward networks (Bengio et al., 2003a) or recurrent networks (Mikolov et al., 2010). Recently auto-regressive versions of transformers have been achieving the best performance on standard benchmarks (Al-Rfou et al., 2019; Dai et al., 2019; Baevski & Auli, 2019). In this section, we describe several specificities of these models that we borrow to make our model work on language modeling, especially with a large vocabulary and a long context.

**Relative position embeddings and caching.** The relative position embeddings are learnable vectors $\mathbf{u}_i$ that are encoding the relative positions in the sequence by setting $\mathbf{p}(t, c) = \mathbf{u}_{t-c}$ in Eq. 3. They replace the fixed absolute position embeddings of the original transformer to allow these models to work on unbounded sequences. When the input sequence is processed in small blocks for efficiency, caching mechanism (Dai et al., 2019) is necessary to ensure that every token $t$ has the same context length regardless of its position in the block.

**Adaptive context size.** In adaptive attention span (Sukhbaatar et al., 2019), each attention head separately learns its context size from data. This allows few heads to have a very long attention span, while others to focus only on recent past. As a result, it becomes possible to extend the maximum attention span without increasing memory footprint and computation time significantly. The method works by multiplying the attention weights in Eq. 4 by a soft-masking function $m_z(t - r)$ that maps values to $[0, 1]$. The real parameter $z \in [0, T]$ controls how much of the attention stays the same, and it is learned together with the rest of the model. Since our attention weights in Eq. 13 contain additional values corresponding to the persistent vectors, we simply pad the masking function with 1 on the locations corresponding to those persistent vectors. This ensures that we only adapt the context size, while the persistent vectors are always included in the attention.

**Adaptive input and output.** In word level language modeling, the size of the vocabulary is very large, making the use of a softmax loss function prohibitive both in terms of running time and memory footprint. A standard solution to circumvent this issue is to replace the full softmax function by the adaptive softmax of Grave et al. (2017a). The idea of the adaptive softmax is to split the vocabulary into disjoint clusters and compare words only within the same cluster. The clusters $\mathcal{V}_1, \ldots, \mathcal{V}_K$ are formed by partitioning the vocabulary $\mathcal{V}$ by following word frequency. The most frequent words are in the first cluster $\mathcal{V}_1$ while the least frequent ones are in the last cluster. The size of each cluster is picked to minimize the overall running time, leading to small clusters of frequent words and large clusters of infrequent words. Finally, they further reduce the running time and the memory footprint by adapting the capacity of the classifiers according to their cluster assignment: The words in the $k$-th cluster have a classifier that is $4^k$ smaller than the one in the first cluster. The underlying motivation

is that infrequent words are hard to predict and there is thus no need to use many parameters for them. The memory footprint of the model is further reduced by tying up the embedding weights with the classifier weights (Inan et al., 2017; Press & Wolf, 2017). In the case of the adaptive softmax, this leads to a special form of embeddings called adaptive input (Baevski & Auli, 2019).

## 5 EXPERIMENTS

### 5.1 EXPERIMENTAL SETUP

In this section, we describe our hyperparameters choices, our optimization scheme as well as the details of the datasets we consider.

**Implementation details.** We initialize token and position embeddings from $\mathcal{N}(0, 1)$, and the matrices $\mathbf{W}_{q,k,v,o}$ from $\mathcal{U}(-\sqrt{d}, \sqrt{d})$. The position embeddings are shared accross all the heads. Persistent vectors are reparameterized by $\mathbf{k}_i = \sqrt{d_h}\mathbf{k}'_i$ and $\mathbf{v}_i = \sqrt{N}\mathbf{v}'_i$, where the parameters $\mathbf{k}'_i$ and $\mathbf{v}'_i$ are initialized from $\mathcal{N}(0, 1/d_h)$ and $\mathcal{N}(0, 1/N)$ respectively. This way the persistent vectors have the same unit variance as the context vectors initially, while the underlying parameters $\mathbf{k}'_i$ and $\mathbf{v}'_i$ are initialized similar to the weights of a feedforward sublayer.

For character level language modeling, we set the model dimension to $d = 512$, and the number of heads to 8. Our small (large) models have 18 (36) all-attention layers, $N = 1024$ (2048) persistent vectors and a dropout rate of 0.3 (0.4) applied to attention weights. The adaptive span has the same hyperparameters as Sukhbaatar et al. (2019) with a maximum span of 8192, except the loss coefficient is set to $10^{-7}$. We use Adagrad (Duchi et al., 2011) with a learning rate of 0.07. We clip individual gradients with a norm larger than 0.03 (Pascanu et al., 2013). We warmup the learning rate linearly for 32k timesteps (Vaswani et al., 2017). A training batch consists of 64 samples, each with 512 consecutive tokens. When the loss on validation stops decreasing, we divide the learning rate by 10 for an additional 20-30k steps. Training large models takes about a day on 64 V100 GPUs.

For word level language modeling, we use a model with $d = 512$ and 36 layers, each with 8 heads and 2048 persistent vectors. We use Adam with a learning rate of 0.00025 and 8k warmup steps. The whole gradient norm is clipped at 1. A batch consists of 64 samples, each with 256 tokens. We use an adaptive span of 2048 with a loss of $5 \times 10^{-7}$. The dropout rate is set to 0.3 for attention weights, and 0.1 for input embeddings and the final representation.

**Datasets and metrics.** For character level language modeling, we consider the `enwik8` and `text8` datasets from Mahoney (2011). Both datasets have a training set of 100M tokens and a vocabulary of 28 and 205 unique characters respectively (including the end-of-sentence token). Both datasets are made of Wikipedia articles split at the character level. The `text8` dataset is preprocessed by lowering casing and retaining only whitespaces and the letters that are in the ISO basic Latin alphabet. We report bit per character (bpc) on dev and test sets.

For word level language modeling, we consider the `WikiText-103` dataset introduced by Merity et al. (2017). The training set of `WikiText-103` contains around 100M tokens and a vocabulary of about 260k words. Each word in the vocabulary appears at least 3 times in the training data. The dataset is made of Wikipedia articles. We report perplexity (ppl) on the dev and test sets.

**Dataset specific implementation details.** Following Baevski & Auli (2019) on `WikiText-103`, we use tied adaptive softmax and adaptive input with 3 clusters of size 20k, 40k and 200k. The dimensions of the classifiers in each cluster are consecutively divided by 4, leading to the following dimensions $d$, $d/4$ and $d/16$.

### 5.2 MAIN RESULTS

We compare our approach to the state of the art on several standard benchmarks on both word level and character level language modeling.

**Character level language modeling.** In Table 1, we report the results on `enwik8`. Our small model outperforms all other models of similar sizes. Our large model matches the state-of-the-art performance with significantly fewer parameters. On `text8`, our small model also matches the best performing model from Sukhbaatar et al. (2019) as shown in Table 2. Our large model is 0.01 bpc

Table 1: Comparison with the state of the art on character level language modeling on `enwik8`. We report bpc for the test set as well as the number of parameters.

| Model | #Params | test bpc |
|---|---|---|
| *Small models* | | |
| Ha et al. (2017) – LN HyperNetworks | 27M | 1.34 |
| Chung et al. (2017) – LN HM-LSTM | 35M | 1.32 |
| Zilly et al. (2017) – Recurrent highway networks | 46M | 1.27 |
| Mujika et al. (2017) – Large FS-LSTM-4 | 47M | 1.25 |
| Krause et al. (2017) – Large mLSTM | 46M | 1.24 |
| Al-Rfou et al. (2019) – T12 | 44M | 1.11 |
| Dai et al. (2019) – Transformer-XL | 41M | 1.06 |
| Sukhbaatar et al. (2019) - Transformer + adaptive span | 39M | 1.02 |
| All-attention network + adaptive span | 39M | **1.01** |
| *Large models* | | |
| Al-Rfou et al. (2019) – T64 | 235M | 1.06 |
| Dai et al. (2019) – Transformer-XL 18l | 88M | 1.03 |
| Dai et al. (2019) – Transformer-XL 24l | 277M | 0.99 |
| Child et al. (2019) – Sparse Transformer (fixed) | 95M | 0.99 |
| Sukhbaatar et al. (2019) - Transformer + adaptive span | 209M | **0.98** |
| All-attention network + adaptive span | 114M | **0.98** |

Table 2: Comparison with the state of the art on character level language modeling on `text8`. We report bpc for the dev and test sets as well as the number of parameters.

| Model | #Params | dev bpc | test bpc |
|---|---|---|---|
| *Small models* | | | |
| Chung et al. (2017) – LN HM-LSTM | 35M | - | 1.29 |
| Zilly et al. (2017) – Recurrent highway networks | 45M | - | 1.27 |
| Krause et al. (2017) – Large mLSTM | 45M | - | 1.27 |
| Al-Rfou et al. (2019) – T12 | 44M | - | 1.18 |
| Sukhbaatar et al. (2019) - Transformer + adaptive span | 38M | 1.05 | **1.11** |
| All-attention network + adaptive span | 38M | 1.05 | **1.11** |
| *Large models* | | | |
| Al-Rfou et al. (2019) – T64 | 235M | 1.06 | 1.13 |
| Dai et al. (2019) – Transformer-XL | 277M | - | 1.08 |
| Sukhbaatar et al. (2019) - Transformer + adaptive span | 209M | 1.01 | **1.07** |
| Transformer + adaptive span | 116M | 1.02 | 1.08 |
| All-attention network + adaptive span | 114M | 1.02 | 1.08 |

below the state-of-the-art, but it matches the performance of a "Transformer + adaptive span" baseline [1] of a similar size.

**Word level language modeling.** In Table 3, we compare the all-attention network with the state of the art among small models on the `WikiText-103` dataset. Our network is 3.4 ppl better than the previous best, which was a Transformer-XL of a comparable size [2]. For completeness, we also report the state of the art obtained with larger models, that is about 2 perplexity points better than us. In Appendix A, we show sample attention maps from our model.

### 5.3 ABLATION STUDY

In this section, we compare different variations of our large model on character level language modeling on `Text8`. First, we vary the number of persistent vectors $N$ in each layer as shown in

---

[1]This baseline has 22 layers, 512 hidden units and 4096 ReLU units.

[2]See Appendix B for Transformer baselines trained using the same code and settings as our model.

Table 3: Comparison with the state of the art on word level language modeling on `WikiText-103`. We report perplexity (ppl) for the dev and test sets as well as the number of parameters.

| Model | #Params | dev ppl | test ppl |
|---|---|---|---|
| *Small models* | | | |
| Grave et al. (2017b) – LSTM | - | - | 48.7 |
| Bai et al. (2018) – TCN | - | - | 45.2 |
| Dauphin et al. (2017) – GCNN-8 | - | - | 44.9 |
| Grave et al. (2017b) – LSTM + Neural cache | - | - | 40.8 |
| Merity et al. (2018) – 4-layer QRNN | 151M | 32.0 | 33.0 |
| Rae et al. (2018) – LSTM + Hebbian + Cache | - | 29.7 | 29.9 |
| Dai et al. (2019) – Transformer-XL Standard | 151M | 23.1 | 24.0 |
| All-attention network + adaptive span | 133M | 19.7 | **20.6** |
| Best published result with a large model (Dai et al., 2019) | 257M | 17.7 | **18.3** |

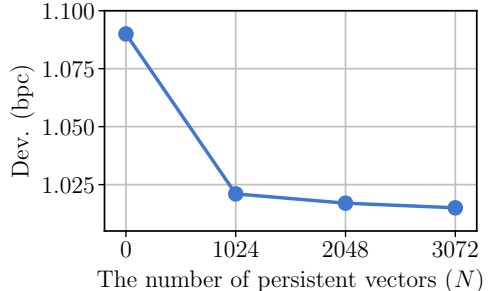 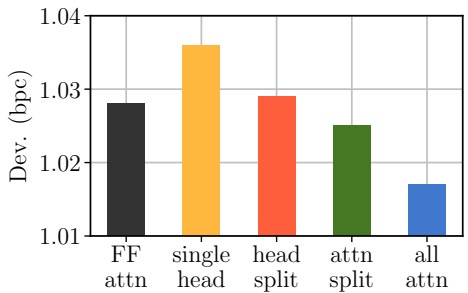

Figure 2: The performance of our large model on `Text8` as we vary **(left)** the number of persistent vectors, or **(right)** the way how persistent vectors integrate with self-attention.

Figure 2(left). The result shows that persistent vectors are crucial for performance, already reaching a good performance at $N = 1024$. A model without persistent vectors (i.e. $N = 0$) is equivalent to a transformer model without feedforward sublayers, and it performs poorly. This also demonstrates the importance of feedforward layers in transformer models. However, it maintains decent performances because it still has a lot of parameters (38M) in the $\mathbf{W}_{q,k,v,o}$ matrices.

We also compare several different ways of integrating persistent vectors into self-attention:

- **All-attn**: this is our default model presented in Section 4 where persistent vectors are simply concatenated to context vectors.

- **Attn-split**: this is the same as "all-attn" except the attention over context and persistent vectors are computed separately. In other words, we replace the softmax in Eq. 13 with two separate softmax functions: one for context vectors only and one for persistent vectors only.

- **Head-split**: this is the same as "all-attn" except we constrain half of the heads to attend only to context vectors, and the other half to attend only to persistent vectors.

- **Single-head**: this is the same as "attn-split", but now persistent vectors are not split into multiple heads. Instead, each layer has a single set of persistent key-value vectors of a dimension $d$.

- **FF-attn**: a Transformer model where the `ReLU` of feedforward sublayers is replaced with a `Softmax` function as discussed in Section 4.1. This is the same as "single-head" above except persistent vectors are kept as a separate sublayer that comes after a self-attention sublayer. Since this will double the depth of a model, we decrease the number of layers to 24 and increase the feedforward size to 3072 to maintain the number of parameters same.

Note that all those versions have the same number of parameters except "head-split", which has fewer parameters because half of its persistent vectors are not used. The result is shown in Figure 2(right). There are few things to notice: **(i)** "all-attn" outperforms "attn-split", which indicates that there is a

benefit in computing attention jointly over persistent and context vectors; **(ii)** "single-head" is worse than "attn-split", which means persistent vectors with more heads are better; and **(iii)** dividing the heads into context-only and persistent-only groups does not work well; and **(iv)** "FF-attn" does not work as good as "all-attn" which means the switch from `ReLU` to `Softmax` alone is not sufficient.

## 6    CONCLUSION

In this paper, we propose a novel attention layer that presents a unified mechanism to aggregate general and contextual information. It extends the self-attention layer of a transformer with a set of persistent vectors that are capable of storing information that is complementary to the short term information in contexts. We also show that these persistent vectors can replace the feedforward layers in a transformer network with no loss of performance. We think that this simplified layer can help better understand how information is processed and stored in transformer-like sequence models.

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

# A    ATTENTION MAPS

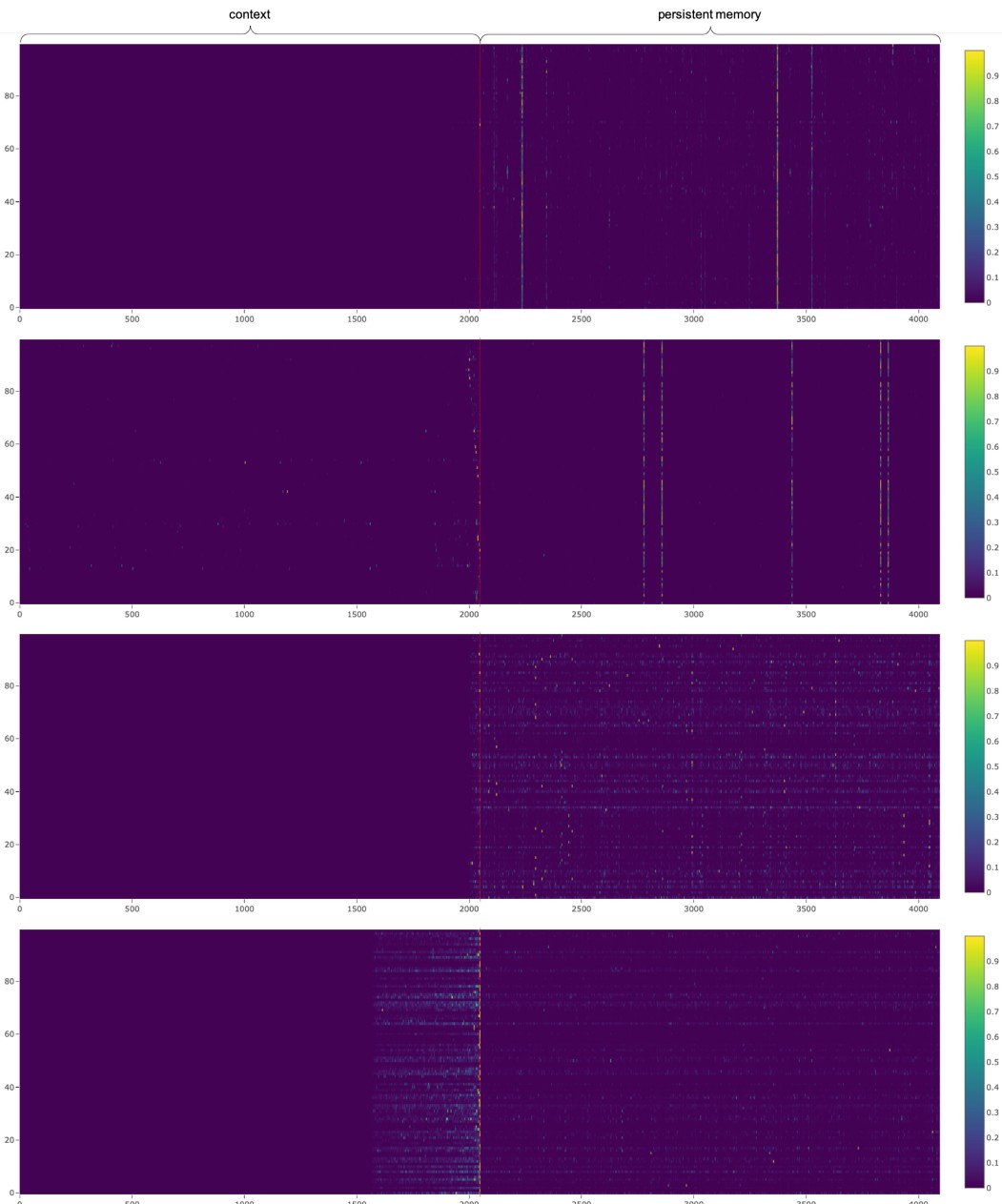

Figure 3: Sample attention maps from our model that trained on the `WikiText-103` dataset. The 4 plots correspond to 4 different attention heads in the model. The $Y$-axis is different samples from a short sequence, and the $X$-axis shows all the vectors in the attention. The first 2048 vectors come from the context, and the remaining 2048 are persistent vectors. In the top 2 heads, few persistent vectors are dominating the attention, although the 2nd head has some attention weights in the context part as well. The 3rd head has more diverse activations on the persistent vectors, while also attending to very recent context. The last head is mostly attending to about last 500 tokens in the context, but there are some activations in the persistent vectors.

# B  BASELINE TRAINING

We trained baseline Transformer models on the `WikiText-103` dataset using the same code and settings as our model to check if some of the training details (e.g. weight initialization, position embeddings, adaptive span, etc.) affected our result in Table 3. We considered two baselines with roughtly the same number of parameters as our model: 1) a 22-layer model with a hidden size of 512 and a feedforward layer size of 4096; 2) a 36-layer model with a hidden size of 512 and a feedforward layer size of 2048. The 22-layer model is more comparable to our model because it has a similar number of nonlinear layers as our model, while 36-layer has twice as much non-linear layers as our model. The training of those models are plotted in Figure 4 against our model (excluding the finetuning part). As we can see, the training of the 36-layer model has diverged early in the training, which is not surprising as such deep Transformer models are known to be unstable during training. The same thing happenned to the 22-layer model but near the end of its training. However, it is almost certain that its final performance would have been worse than our model even if its training did not diverge.

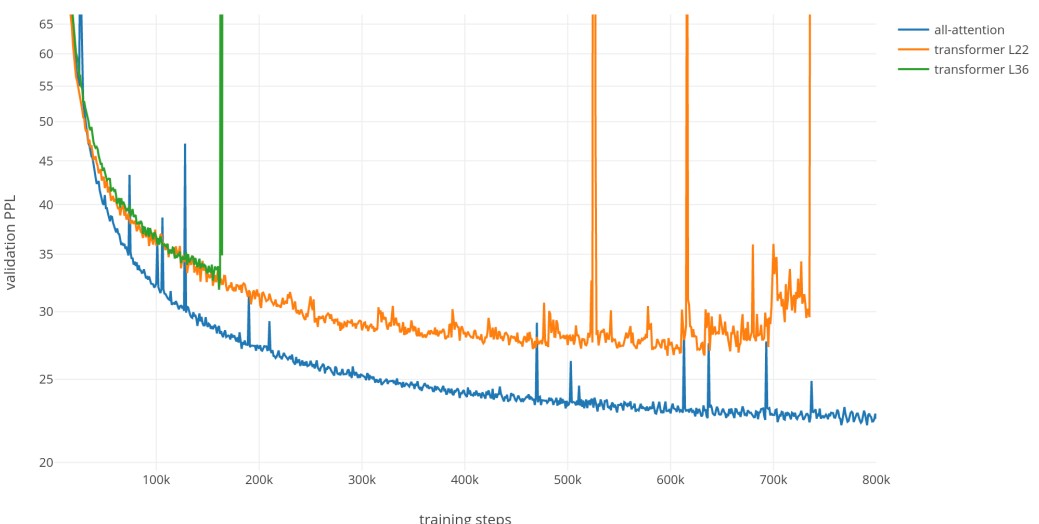

Figure 4: Training of baseline Transformer models on `WikiText-103` dataset.

