# OpenReview forum: "Augmenting Self-attention with Persistent Memory"
_ICLR.cc/2020/Conference — Reject_

### Official Review · AnonReviewer3 · 2019-10-07
**Official Blind Review #3**

**Rating:** 3

**Review:**

This paper proposes a simple modification to the ubiquitous Transformer model. Noticing that the feed-forward layer of a Transformer layer looks a bit like an attention over "persistent" memory vectors, the authors propose to explicitly incorporate this notion directly into the self-attention layer. This involves concatenating the contextual representations with global, learned memory vectors, which are attended over.

The model is tested on widedly-utilized character/word-level language modeling benchmarks, where it is found to outperform, or be on par with, existing models while using fewer number of parameters.

Insofar as architectural advancements can translate to general improvements across multiple NLP tasks, this paper could be seen as important. However, I am not sure that in 2019, demonstrating arguably-marginal perplexity improvements on standard datasets is enough. I would love to see if this type of layer can result in better conditional generation models (e.g. translation, summarization), or can train GPT2/BERT/XLNet-style models whose representations better transfer to other tasks.

I had some further questions/comments:

- I found the motivation of the persistent memory vector as replacing the FF-layer somewhat tenuous. Eq(5) is definitely different from Eq(9)! In my opinion this work can be better motivated/presented as just a standalone modification to the Transformer layer.

- While it is impressive that the proposed approach performs better (or on par with, in the case of character-level language modeling) than the previous state-of-the-art models which are larger, to me it is not immediately clear if the benefit is coming from the proposed modifications, or something else (e.g. some of the things mentioned in 4.3). I understand most of this is taking from prior work, but as we all know, in deep models various architectural/hyperparameter modifications can interact in unexpected ways. Therefore, at a minimum, I would like to see the performance of a comparable model with the same exact setting, except for the persistent attention layer (i.e. N = 0 but using the feedforward sublayers). (If I understand the work correctly, this baseline should have comparable number of parameters?)

- The ablation studies are good but it would be good to see them on the word-level task as well.

- What is the performance of the FF-attn baseline with the same depth? (I understand the number of parameters would be  larger, but does this model perform as well as the Dai et al. 2019 work?)

- What if you combine this with the FF sublayer as well?

- Have you tried qualitatively analyzing the attention distributions? What are some examples in which the persistent memory vectors are attended to most? What are some examples in which the attention distribution for this model differs considerly versus regular self-attention layers?

[EDIT after author rebuttal]
Thank you very much for the rebuttal. I have updated my score to reflect the latest iteration of the paper.

**Experience Assessment:**

I have published one or two papers in this area.

**Review Assessment: Checking Correctness Of Derivations And Theory:**

I carefully checked the derivations and theory.

**Review Assessment: Checking Correctness Of Experiments:**

I carefully checked the experiments.

**Review Assessment: Thoroughness In Paper Reading:**

I read the paper at least twice and used my best judgement in assessing the paper.

---

> ### Author Response · Authors · 2019-11-15
> **Response to the review**
>
> Thank you for taking time to review our paper and the useful suggestions. However, we have to respectfully disagree with the reasoning behind the rejection. We agree that having experiments like GPT2/BERT/XLNet-style would make our paper stronger, but this shouldn’t be a reason for rejection. Training such large-scale experiments often take lot of efforts and resources that goes beyond the scope of this paper, and also unlikely to add any more insights to our approach.
>
> The goal of the paper is not to push the state-of-the-art regardless of model size and complexity, but it’s rather about simplifying the current architecture while maintaining its performance. We chose language modeling tasks because it’s at the core of all those tasks, and rich in baselines of various architectures. However, we agree with the point about missing a same-sized baseline, so we added it in the updated version. Our model still outperforms this baseline as well as the similar-sized TransformerXL baseline.
>
> Here are our answers to your questions:
> - “Eq(5) is definitely different from Eq(9)!”
> - Yes, they are not exactly the same. We added section 4.1 to help readers better understand the intuition behind our method, and how we come up with the idea. We modified the introduction to make our motivation more clearer.
>
> - “in deep models various architectural/hyperparameter modifications can interact in unexpected ways. Therefore, at a minimum, I would like to see the performance of a comparable model with the same exact setting”
> - This is a great point and we agree that training details can affect the model performance. Therefore, as suggested, we added a comparable baseline Transformer trained with our code using exactly the same setting as our model in the appendix B. Although the training diverged in near the end (deep transformer models are known to be unstable during training), we can clearly see its performance is worse than our model.
>
> - “What is the performance of the FF-attn baseline with the same depth?”
> - Comparing the depth of two different architectures is a little problematic. A transformer layer actually consists of two sublayers, so its depth could be viewed as 2. In that case we added a baseline with 44 sublayers in the appendix B as mentioned above. If the depth is the number of transformer layers, then a same-depth baseline would have 36 layers, or 72 sublayers. This means it has twice as many nonlinear layers as our model. Training a such deep transformer model is known to be very unstable. We did try to train it, but the training diverged after 157k updates as shown in the appendix B. We tried to make it more stable by reducing the gradient clipping without success.
>
> - “What if you combine this with the FF sublayer as well?”
> - Combining FF sublayer is an interesting idea that might improve the performance, but as mentioned above, the point of the paper was to simplify the architecture using a single attention mechanism, rather than pushing the limits of the SOTA.
>
> - “Have you tried qualitatively analyzing the attention distributions?”
> - This is a great question. We have tried looking at persistent memory attention qualitatively, but it’s challenging to find any meaningful persistent vector because there are more than half a million of them. Plus, unlike context vectors, they lack any temporal structure or direct connections to input tokens. However, we have included sample attention maps in the appendix A, where we observed several different types of patterns. The attention over persistent vectors is definitely different from the one over context vectors. First they look very random, because they don’t have any structure. Also, we observed that sometimes few persistent vectors would dominate the attention, which could be a focus of a follow-up work.
>
> - “The ablation studies are good but it would be good to see them on the word-level task as well”
> - Yes, it would be better to have all the ablation studies on all the datasets. However, considering that even training a single word-level model takes a significant amount of resources, we decided to only perform the ablation study on a single dataset.

---

### Official Review · AnonReviewer2 · 2019-10-22
**Official Blind Review #2**

**Rating:** 6

**Review:**

This paper considers an architecture change to the transformer in which they swap the feedforward subcomponent of the standard transformer with an "attention only" variant that includes persistent "memory" vectors. The model is evaluated against a suite of baselines on the tasks of character- and word-level language modeling. Combining this "all attention" approach with adaptive span yields results about equivalent to the SOTA, in some cases with fewer parameters than existing models. The authors do a nice job of presenting ablation results. A key finding here, for example, is that the a model stripped of both persistent vectors and the feedforward sublayer performs poorly.

Overall, I'm on the fence regarding this submission. This is solid work, and the idea of exploiting persistent representations in the transformer seems promising. But the architecture change here is relatively minor, and the gains seem somewhat minor (the exception may be in Table 2 which shows equivalent performance with half the parameters to previous SOTA, but then no other model is in the range of O(100m) parameters, so hard to know what's going on here).

One thing I would have liked is more motivation. If equivalence with fewer parameters is the main aim, then the model seems to fair reasonably well but the results are not really compelling. If, on the other hand, the authors are primarily interested in exploiting persistence, then I think this could have been investigated a bit more exhaustively, and perhaps the focus need not be on necessarily replacing the feedforward subcomponent (although that is one reasonable strategy). I do not see a huge inherent advantage to removing the feedforward layer, and it seems like there are alternative strategies --- at least equally as good --- to reduce parameters.

A question for the authors: did you consider a vanilla Transformer with persistent memory vectors added? This would be something like adding a constant dummy input (independent of the example) that would be passed forward and arbitrarily transformed. I guess the meta-point here is that it doesn't seem to me that the persistent representations and the feedforward sublayer are necessarily mutually exclusive.

As a minor comment, I think Eq 13 is redundant since it literally repeats Eq. 4 save for swapping in $C_t^+$ for $C_t$.

**Experience Assessment:**

I have published one or two papers in this area.

**Review Assessment: Checking Correctness Of Derivations And Theory:**

I assessed the sensibility of the derivations and theory.

**Review Assessment: Checking Correctness Of Experiments:**

I assessed the sensibility of the experiments.

**Review Assessment: Thoroughness In Paper Reading:**

I read the paper at least twice and used my best judgement in assessing the paper.

---

> ### Author Response · Authors · 2019-11-15
> **Response to the review**
>
> Thank you for the constructive and insightful review. Find below our responses to your questions:
>
> - “no other model is in the range of O(100m) parameters”:
> - This is true. So we added a “Transformer + adaptive-span” baseline with a similar number of parameters as our model. It matched the performance of our model, confirming that our simplification has not degraded the performance. However, we would like to point out that we have a similar sized baseline in Table 3, which our model outperformed by a large margin (in appendix B, we also added baselines of a similar size using our implementation to separate out the improvements that could have brought by some of the training details).
>
> - “I would have liked is more motivation”
> - We agree that our motivation might not have been conveyed clearly enough in the manuscript, so we tried to make it clearer in the updated version. Our motivation was to simplify the Transformer architecture without degrading its performance. We hope that this simplification will lead to new insights and better analysis of the model. But the smaller number of parameters in the word-level LM task is a concrete outcome that, by itself, could justify our approach.
>
> - “adding a constant dummy input (independent of the example) that would be passed forward and arbitrarily transformed”:
> - This is definitely an interesting idea to explore. However, it is quite different and orthogonal to our approach because 1) they are transformed at every layer and 2) have learnable parameters only at the embedding layer. Because of 2, such a model will require a very large number of dummy inputs to have a comparable number of parameters. Because of 1, it would be very slow to train because the dummy inputs have to be re-transformed at every step, thus significantly reducing the parallelism of Transformer models.

---

### Official Review · AnonReviewer1 · 2019-10-25
**Official Blind Review #1**

**Rating:** 6

**Review:**



This paper mainly proposes a modification of well-studied Transformer architecture that is widely used for the text generation tasks, i.e., language modeling and machine translation.
The main idea is to consist of Transformer architecture only with self-attention layers. In other words, the proposed method discards the feed-forward layers and augment the self-attention layers with persistent memory vectors.
They conduct experiments on character and word-based language modeling and show the on-par or slightly better performance comparing with the standard Transformer language model and other similar Transformer modifications.

The proposed method is just a modification of the existing neural network architecture.
Moreover, their method did not significantly improve the performance of language modeling in the experiments.
From these perspectives, the proposed method is not innovative.

However, the Transformer architecture is currently applied to a wide variety of tasks in text generation. Therefore, the proposed method can be largely influential to the community.
Actually, I like an idea discarding the feed-forward (sub-)layers in the Transformer, whose intuitive (and theoretical) role is not much discussed in the literature.



**Experience Assessment:**

I have published in this field for several years.

**Review Assessment: Checking Correctness Of Derivations And Theory:**

I carefully checked the derivations and theory.

**Review Assessment: Checking Correctness Of Experiments:**

I carefully checked the experiments.

**Review Assessment: Thoroughness In Paper Reading:**

I read the paper thoroughly.

---

### Decision · Program_Chairs · 2019-12-19

**Decision:**

Reject

**Comment:**

This paper proposes a modification to the Transformer architecture in which the self-attention and feed-forward layer are merged into a self-attention layer with "persistent" memory vectors. This involves concatenating the contextual representations with global, learned memory vectors, which are attended over. Experiments show slight gains in character and word-level language modeling benchmarks.

While the proposed architectural changes are interesting, they are also rather minor and had a small impact in performance and in number of model parameters. The motivation of the persistent memory vector as replacing the FF-layer is a bit tenuous since Eqs 5 and 9 are substantially different. Overall the contribution seems a bit thin for a ICLR paper. I suggest more analysis and possibly experimentation in other tasks in a future iteration of this paper.